

# Differences in external load among indoor and beach volleyball players during elite matches

Mikulas Hank[1], Lee Cabell[2], Frantisek Zahalka[1], Petr Miřátský[1], Bohuslav Cabrnoch[1], Lucia Mala[1] and Tomas Maly[1]

[1] Sport Research Center, Charles University, Faculty of Physical Education and Sport, Prague, Czech Republic
[2] Department of Physical Therapy, Carroll University, Waukesha, United States

## ABSTRACT

The aim of this cross-sectional study was to examine relationships of external load variables between beach and indoor volleyball amongst individual positions on the team. The movements of eight beach and fourteen indoor female volleyball players were recorded during elite playoff matches; in total, 2,336 three-dimensional trajectories were analyzed. Time-outs and intervals between rallies or sets were excluded from active play time. In both beach and indoor volleyball, 80% of rallies lasted up to 10 s, and players covered 4.5 to 10 m of court during 60% of rally play. Differences in dependent variables of external load were found between independent variables of sports and player positions ($p < 0.05$). The distance covered in beach volleyball rallies and Player Load™ parameters was significantly higher by up to 23%. The unstable court surface with sand in beach volleyball elevated explosive Player Load™ (accelerations in all three orthogonal planes of motion higher than 3.5 m/s$^3$) in beach volleyball players compared to those of players on stable flooring in indoor. While beach volleyball blocker and defender positions showed no significant difference in parameters between each other, they differed in all parameters when compared to player positions in indoor volleyball. Indoor blocker and libero reached higher loads than setter, outside and opposite positions in various parameters. Factors that influence external load include the larger relative court areas covered by each player in beach volleyball, complexity of players' roles, and game strategy. This data adds to the knowledge of elite match demands in female volleyball. Specified agility-drill distances and times are essential for training optimization and must be supported by scientific observation. Researchers, coaches, and conditioning specialists should find this helpful for achieving a higher degree of training regulation.

# INTRODUCTION

External load (EL) analysis during competition is crucial to understand the demands for match play but is rarely studied in women's volleyball (*João et al., 2021*). Video recording of EL in volleyball and between player positions was mostly used in jump parameters

Corresponding author
Mikulas Hank,
miki.hank@gmail.com

evaluations and in males volleyball players (*Pisa et al., 2022*; *Sheppard, Gabbett & Stanganelli, 2009*). However, EL derived from active playing time (individual rallies) such as distance covered across the court and the level of acceleration, deceleration and change in direction is lacking in women's volleyball for both indoor (IV) and beach (BV). When compared to total distance covered alone (*Hader et al., 2019*), performance analysis in sports relies often on relative distance in specific acceleration zones (*Bellinger et al., 2021*; *Vlantes & Readdy, 2017*), or jump count/height (*Pisa et al., 2022*), especially because more time spent in higher zones put higher metabolic load on players, as it is linked to acceleration/deceleration movement, rapid change of direction and other motions considered to be explosive (*Altundag et al., 2022*). The physical demands for an athlete in training or competition may be influenced by the sports' rules time spent in activity *vs* rest, the number of players, differences in court size, duration of competition, roles of player positions, or surface type where the activity is performed (*Balasas et al., 2018*; *Jerome et al., 2023*; *Palao, Manzanares & Valades, 2014*; *Smith, 2006*; *Wang & Yuan, 2022*). Indoor and beach volleyball consist of peak power-related movements where biomotor abilities of speed and strength are required for spiking, maximizing jump height, transitioning quickly, recovering from falls, and isometric low stances (*Sheppard, Gabbett & Stanganelli, 2009*). Matches and training sessions are composed of these intermittent physical demands, often described as a load (*Pisa et al., 2022*). While EL can be represented by time duration, movement distance, speed, jump count, or jump height (*Pelzer et al., 2020*), internal load (IL) reflects the biological responses to EL, such as heart rate (*Cressey et al., 2007*), cardiovascular performance, or blood lactate concentration (*Lupo, Ungureanu & Brustio, 2020*). The relationship between IL and EL seems to be positively associated, as demonstrated by *McLaren et al. (2018)* found moderate to large (r = 0.48 to 0.82) ties between heart rate and perceived-exertion internal measures with EL data. This is based on distance covered or other accelerometer data. Comparisons in IL and EL between player positions have not been well investigated (*Palao et al., 2015*). Quantification of match IL and EL is crucial for training designs, training optimization, and player individualization in sports games (*Clemente et al., 2019*; *Oliveira et al., 2018*). Training optimization may lead to injury reduction (*Visnes & Bahr, 2013*), prevent over/undertraining (*Gabbett, 2020*) and boost individual or team performance (*Clemente et al., 2019*). Various approaches have been used to evaluate external load. Microsensor technology with Global Positioning System (GPS) is used to obtain positional data in sports, including BV (*João et al., 2021*), but is not always possible to measure during competition (*Vlantes & Readdy, 2017*). GPS sensors are widely used outdoors, but cannot be used indoors, and they are less reliable in high-intensity movements due to lower sample rates (*Nicolella et al., 2018*; *Rico-González et al., 2020*). Wearable accelerometers measure acceleration-based thresholds like inertial movement analysis but not information on an athlete's position (IMA) (*Nicolella et al., 2018*). Besides the higher sample rate (100 Hz), it decreases reliability in relation to rising movement speed and "multiplane, high-intensity actions" (*Nicolella et al., 2018*; *Wundersitz et al., 2015*). One of the IMA parameters is Player Load™ (PL) (Catapult Innovations, Melbourne, Australia). Authors are not united in the definition, calculation, nor methodological approach to evaluate the PL (*Bredt et al., 2020*). In general, it is

calculated by recording accelerations from three different planes using an accelerometer and is often used to quantify training or EL in elite sports (*Wik, Luteberget & Spencer, 2017*). During physical activity, PL value rise with increasing frequency of accelerations. We may understand PL as the sum of rates of change in acceleration, or jerks (*Nicolella et al., 2018*). The parameter PL was described as follows: "a modified vector magnitude, expressed as the square root of the sum of the squared rates of change in acceleration between each moment of a training session in each movement axis (x, y, and z)" (*Bredt et al., 2020*). PL becomes a used parameter and a part of determining EL in sports (*Schelling & Torres, 2016*), especially in explosive sports like volleyball, where sum of jerks may represent values of acceleration load (*Vlantes & Readdy, 2017*). However, researchers have noticed its limitations as PL has no direct relation to the acceleration's magnitude and past research is divided in the use of various non-standardized equations. Researchers also did not present PL formula or interpretation for PL (*Bredt et al., 2020*). Used technology, appropriate description of methodology, calculation formula and appropriate units ($m/s^3$) have to be described within the research for reproducibility cases (*Bredt et al., 2020*).

To obtain positional data in real time and calculate accelerations when it is not allowed to wear sensors in IV and BV matches, multi-camera video recording may by the next approach primarily in an indoor environment where GPS sensors are not working. Video recording is less invasive during EL data collection, while often needs time-consuming post-processing when tracking objects (athletes' positions) (*Nicolella et al., 2018*). From the point of space requirements and relative player area in IV, six players share a 9 × 9 m (81 $m^2$; 13.5 $m^2$/per player) court with a solid/stable surface, whereas in BV two players share a smaller court (8 × 8 m) (64 $m^2$; 32 $m^2$/per player) with a deformed, unstable surface. Playing surface affects physiological and biomechanical loads and causes various adaptations (*Binnie et al., 2014*). Unstable surface like sand may cause increased activity of the lower limb muscles and stabilization in hip, knee, and ankle joints, but also lower propulsion during force production (*Giatsis, Panoutsakopoulos & Kollias, 2018*).

The energy athletes expend may rise up to 3.7 times from firm to unstable surfaces, but the level of impact is reduced (*Binnie et al., 2014*) while firm surfaces are associated with an increased number of injuries and overuse related to impact (*Ekstrand, Timpka & Hagglund, 2006*). Therefore, sand offers a unique stimulus, but also a viable option for recovery that improves aerobic and anaerobic fitness (*Impellizzeri et al., 2008*). Running on sand, however, was associated with an increased rate of mid-portion Achilles tendinopathy (*Knobloch, Yoon & Vogt, 2008*), but reports of lower limb injury was lower in BV than IV (*Reeser et al., 2006*).

BV player positions consist of a defender (Defender$_{BV}$), blocker (Blocker$_{BV}$), or no specialization (sharing the roles), while IV involves more specialized positions such as a blocker, setter, libero, outside, and opposite; thus, specific loads are hypothesized (*Marques et al., 2009*). Additionally, BV players tend to be older but more agile and shorter than IV players (*Palao, Gutierrez & Frideres, 2008*). In terms of physical demands determination in volleyball, *Lidor & Ziv (2010)* proposed to examine the IL and EL in match conditions. The analysis of EL in a volleyball match is very important for coaches because they can understand the motor activity of players (*Mroczek et al., 2014*). Movement performance

data during volleyball sports are limited, while recent research also lacks how the active/rally time was standardized, or at what moment and why the beginning or end of a recording was presented (*Bellinger et al., 2021*). The two types of volleyball are played worldwide and have common motor characteristics, but environment, surface type, relative area per player (32 m$^2$ per player in BV, and 13 m$^2$ per player in IV) or player position may significantly characterize EL, resulting in the specific physical demands and player preparation (*Marques et al., 2009*; *Smith, 2006*). Therefore, the aim of this study was to examine the differences in selected EL parameters between two types of volleyball, and between different player positions within standardized active playing time during elite women's matches. It was hypothesized that there would be a difference in the EL and PL parameters between IV and BV. Please note that the difference is not calculated between all IV player positions when compared to the BV player positions. This is due to setter's and libero's constant changes of position across the court during defense and attack assistance (*Marques et al., 2009*; *Mroczek et al., 2014*; *Sheppard, Gabbett & Riggs, 2012*) and each courts' specifications (*Smith, 2006*).

## MATERIALS AND METHODS

### Participants

This is a cross-sectional study. Eight elite female BV players (age = 28.6 ± 5.8 years; height = 178.1 ± 5.7 cm; weight = 66.5 ± 5.3 kg) from the beach volleyball Fédération Internationale de Volleyball (FIVB) World Tour playoff and fourteen elite female IV players (age = 25.2 ± 6.1 years; height = 182.3 ± 6.2 cm; weight = 72.1 ± 5.8 kg) from the indoor volleyball Confederation Européenne de Volleyball (CEV) Champions League playoff match were observed. Players were categorized based on their playing positions to independent variables like Blocker$_{BV}$ ($n = 4$) and Defender$_{BV}$ ($n = 4$) in the BV group, and to Blocker ($n = 4$), Setter ($n = 2$), Libero ($n = 2$), Outside ($n = 4$) and Opposite ($n = 2$) in the IV group. Professional players at this level train up to 25 h per week (*Sheppard et al., 2008*). During data collection, all players were free of injury and illness. We have received written informed consent from participants to the data collection according to Declaration of Helsinki. The research was approved by the independent ethics committee of Faculty of Physical Education and Sport at Charles University under the number EC 259/2020.

### Procedures

Before the study, a power analysis (G*Power 3.1.9.7) (*Faul et al., 2007*) was run to estimate sample sizes for the two volleyball groups. Differences between BV and IV teams in pilot research were used to calculate that for an effect size (ES) of 1.3, with Alpha set as 0.05 and Beta as 0.2, given a power of 0.8, the estimated sample size required for this study was 15 per group. We used 14 top IV athletes and eight BV athletes, which is still not enough but, based on the number of total rallies and movement trajectories, we were able to achieve variable comparison between groups. A total of 256 international BV ($n = 92$ rallies; four sets) and IV ($n = 164$ rallies; four sets) matches were videotaped using HD digital camcorders setup in two 90-degree angles in a stationary position (HDC90E; Sony Ltd., Tokyo, Japan; 50 fps; 1,920 × 1,080 pixels). Before the match recordings, two calibration

cubes (1 m × 1 m × 2 m) were placed in back-court corners for BV and IV. For post-processing, space reconstruction and kinematic three-dimensional (3D) analysis of players were performed with TEMA Trackeye v2.3 (Image Systems, Linköping, Sweden). Reconstruction of the spatial coordinates reached average residues of 0.025 ± 0.005 m. Semi-automatic tracking of the center of the head was chosen for the player's position and movement determination.

The beginning and end of each rally was measured from the moment of service hit by the server until the ball was out of play under the conditions of official rally description (the sequence of playing actions) (*de Volleyball, 2016*). The moment the ball was tossed up for a serve and reached the player's head was set as the beginning of the rally, and the moment the ball hit the floor or out area at the net was set as the end of the rally. "*Pseudo rallies*" or straight point or serve mistakes were excluded for this study. A total of 2,336 rally movement trajectories were further processed and analyzed by TEMA Trackeye v2.3. MATLAB® software (The MathWorks, Inc., Natick, MA, USA) was used for calculation of the duration of rally (DR), total rally horizontal distance covered (TD), and relative distance covered during a rally over the time played (rTD). PL from each 20-ms time interval in three-plane (side/side X, up/down Y, forward/backward Z) positional data of players' movement was determined in individual rallies using the cartesian formula presented in the study by *Nicolella et al. (2018)* was used as follows:

$$PL = \sqrt{\frac{\left(a_{y(t)} - a_{y(t-1)}\right)^2 + \left(a_{x(t)} - a_{x(t-1)}\right)^2 + \left(a_{z(t)} - a_{z(t-1)}\right)^2}{100}}.$$

This was subsequently categorized to relative PL (rPL; PL divided by duration of rally in minutes) and explosive PL (ePL) described by *Vlantes & Readdy (2017)* as count of accelerations greater than 3.5 m/s$^2$ in the horizontal plane (mediolateral (x) and anteroposterior (y) axes).

## Statistical analysis

Normal data distribution was tested using the Shapiro-Wilk test, and the homogeneity of variance was tested using Bartlett's test. Percentages and medians with 5[th] and 95[th] percentiles were used for descriptive analysis. Comparisons of EL parameters (dependent variable) between the two independent samples (sports; independed variable) were analyzed using the Mann-Whitney U Test with confidence interval (CI) of 95%. The global (omnibus) comparison of player positions was conducted using the Kruskal-Wallis test. The 95% CI estimate was computed for a medium value. This was followed by the Mann-Whitney U Test for all pairwise comparisons. We report effect sizes (ES) with 95% confidence intervals in form of $\eta^2$ and Cohen's d for omnibus and pairwise comparisons, respectively. It was categorized as follows: <0.19 negligible; 0.20–0.49 small; 0.50–0.79 medium; >0.80 large (*Cohen, 1992*). Graph representation of ES with Cohen's d 95% CI was used according to *Salerno et al. (2023)*. The data were processed using IBM SPSS v25 (Statistical Package for Social Sciences, Inc., Chicago, IL, USA).

## RESULTS

### Comparison of external load parameters between beach and indoor volleyball

Descriptives and pairwise comparison between two sports is shown in Table 1. The longest rally reached 18.8 s in BV and 21.5 s in IV, but rallies which lasted less than 10 s accounted for 83% of BV rallies and 81% of IV rallies. At the same time, 10% of BV and 15% of IV rallies lasted no longer than 4 s. Players in both BV and IV reached between 4.5 to 10.1 m TD in 62.3% of all rallies. Players in the BV group attained significantly higher TD values which differed up to 6.6% with a small effect (Fig. 1). A significant difference of 10.8% with a small effect was also found in rTD. The PL differed by up to 10.5% in favor of BV, while the rPL and ePL were also greater in BV when compared to IV with small effects.

### Comparison of external load parameters based on players' position

Descriptives and global statistics between player positions is shown in Table 2. A player's position revealed a significant effect on rTD ($\chi^2 = 19.336$, df = 6, $p = 0.004$), on PL ($\chi^2 = 35.184$, df = 6, $p < 0.001$), on rPL ($\chi^2 = 81.156$, df = 6, $p < 0.001$), and on ePL ($\chi^2 = 69.768$, df = 6, $p < 0.001$). The pairwise comparison revealed higher values of rTD in BV players with small effect (Fig. 2), when Blocker$_{BV}$ overreached indoor Blocker ($p = 0.013$, d = 0.25, 95% CI [−0.031 to 0.525]), Setter ($p = 0.002$, d = 0.34, 95% CI [0.045–0.632]), Outside ($p < 0.001$, d = 0.33, 95% CI [0.033–0.619]), and Opposite ($p = 0.004$, d = 0.30, 95% CI [0.011–0.595]) by almost 15%. Defender$_{BV}$ showed 8.2% higher rTD when compared to indoor Setter ($p = 0.048$, d = 0.22, 95% CI [−0.074 to 0.507]) and Outside ($p = 0.021$, d = 0.20, 95% CI [−0.088 to 0.493]). Libero reached significant difference in PL by almost 23% when compared to Setter ($p = 0.008$, d = 0.31, 95% CI [−0.041 to 0.663]), Outside ($p = 0.030$, d = 0.20, 95% CI [−0.151 to 0.548]), and Opposite ($p = 0.001$, d = 0.40, 95% CI [0.034–0.743]). Both BV player positions showed higher PL by 21% when compared to Opposite ($p < 0.001$, d = 0.48, 95% CI [0.181–0.782]), Setter ($p < 0.001$, d = 0.40, 95% CI [0.091–0.686]) and Outside ($p < 0.001$, d = 0.27, 95% CI [−0.023 to 0.567]). Within the IV group, the PL of Blocker was higher and significantly different from Setter ($p = 0.034$, d = 0.22, 95% CI [−0.058 to 0.501]) and Opposite ($p = 0.004$, d = 0.31, 95% CI [0.026–0.588]). Eventually, significant difference was found in rPL between Blocker *vs* Opposite ($p < 0.001$, d = 0.56, 95% CI [0.273–0.849]), Outside *vs* Opposite ($p = 0.004$, d = 0.26, 95% CI [−0.050 to 0.561]) and both BV players *vs* Setter ($p < 0.001$, d = 0.60, 95% CI [−0.290 to 0.899]) and Opposite ($p < 0.001$, d = 0.61, 95% CI [−0.305 to 0.916]). Furthermore, Outside reached larger rPL than Setter ($p = 0.008$, d = 0.23, 95% CI [−0.011 to 0.443]), Blocker$_{BV}$ *vs* Outside ($p = 0.002$, d = 0.268, 95% CI [−0.027 to 0.562]), Defender$_{BV}$ *vs* Outside ($p < 0.001$, d = 0.32, 95% CI [0.024–0.616]) and Libero *vs* Outside ($p < 0.001$, d = 0.37, 95% CI [0.012–0.714]) and Opposite ($p < 0.001$, d = 0.715, 95% CI [0.345–1.079]). Highest ePL was reached by Libero when compared to Opposite ($p < 0.001$, d = 0.56, 95% CI [0.196–0.913]), Setter ($p < 0.001$, d = 0.54, 95% CI [−0.184 to 0.900]) and Outside ($p = 0.001$, d = 0.31, 95% CI [−0.044 to 0.655]).

**Table 1** Results represented as medians (5th–95th percentiles) and statistical comparison between parameters of beach and indoor volleyball.

| Variable | Beach volleyball | Indoor volleyball | $\chi^2$ | $p$ | ES (95% CI) |
|---|---|---|---|---|---|
| DR (s) | 6.15 (2.32–15.80) | 5.64 (2.16–18.01) | 2.50 | 0.115 | 0.20 [−0.06 to 0.45] |
| TD (m) | 7.91 (2.01–21.21) | 7.11 (1.54–22.59) | 9.41 | 0.002 | 0.17 [0.05–0.29] |
| rTD (m/min) | 72.62 (33.31–133.86) | 68.58 (28.57–111.59) | 14.10 | <0.001 | 0.20 [0.08–0.32] |
| PL (m/s³) | 231.32 (62.46–570.63) | 196.14 (42.89–605.00) | 19.41 | <0.001 | 0.24 [0.12–0.36] |
| rPL (m/s³/min) | 2,113.09 (1,230.10–3,096.83) | 1,955.20 (865.83–3,094.52) | 21.02 | <0.001 | 0.25 [0.13–0.37] |
| ePL (m/s³) | 193.72 (50.99–492.08) | 155.49 (32.36–547.76) | 28.51 | <0.001 | 0.30 [0.17–0.42] |

Note:
df = 1 for all comparisons; DR, duration of rally; TD, total rally horizontal distance covered; rTD, relative total rally horizontal distance covered TD; PL, player load[TM]; rPL, relative player load[TM]; ePL, explosive player load[TM] (acceleration >3.5 m/s³).

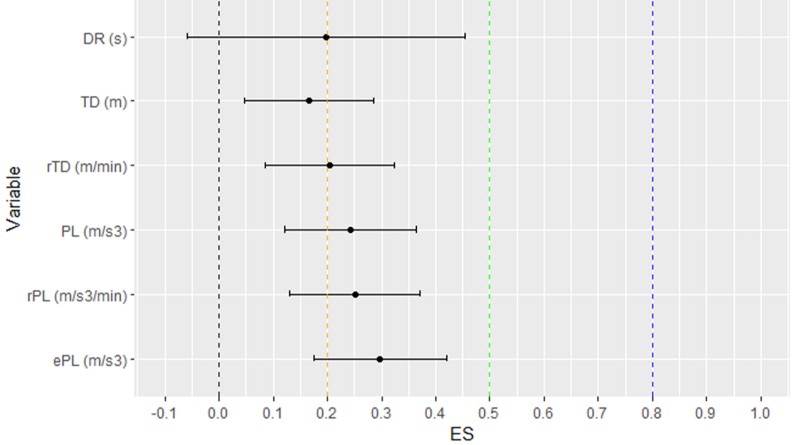

**Figure 1** ES with corresponding confidence intervals for pairwise comparison between sports.

**Table 2** Results represented as medians (5th–95th percentiles) and global statistics $p$-values between parameters of player positions.

| Variable | Blocker$_{BV}$ (a) | Defender$_{BV}$ (b) | Blocker (c) | Setter (d) | Libero (e) | Outside (f) | Opposite (g) | $p$ | ES (95% CI) |
|---|---|---|---|---|---|---|---|---|---|
| TD (m) | 8.23 (1.98–21.24) | 7.82 (1.78–21.21) | 7.73[a] (1.07–21.55) | 6.90[a] (1.68–22.75) | 7.29 (1.54–24.72) | 7.11[a] (1.51–23.00) | 6.76[a] (1.56–21.40) | 0.052 | 0.14 [−0.14 to 0.24] |
| rTD (m/min) | 73.16 (35.66–144.36) | 71.93 (32.00–125.75) | 67.96[a] (23.10–121.46) | 68.55[a, b] (29.56–108.88) | 71.64 (38.31–108.30) | 67.37[a, b] (29.39–109.34) | 68.43[a] (21.71–105.95) | 0.004 | 0.20 [−0.12 to 0.31] |
| PL (m/s³) | 223.43 (63.07–570.63) | 233.10 (54.78–587.55) | 216.96 (27.88–605.83) | 184.31[a, b, c, d] (44.41–509.21) | 208.06 (36.75–695.88) | 190.29[a, b, d] (43.16–618.52) | 162.40[a, b, c, d] (44.70–526.57) | <0.001 | 0.30 [0.17–0.39] |
| rPL (m/s³/min) | 2,081.05 (1,098.66–3,139.72) | 2,160.40 (1,235.88–3,059.24) | 2,155.71 (590.23–3,128.23) | 1,765.49[a, b, c, d, e] (788.39–3,081.43) | 2,211.28 (1,125.17–3,380.19) | 1,956.61[a, b, c, d] (908.23–2,988.76) | 1,776.96[a, b, c, d, e] (876.53–3,079.38) | <0.001 | 0.50 [0.36–0.59] |
| ePL (m/s³) | 191.77 (50.66–468.52) | 195.64 (50.45–505.24) | 178.35 (45.54–564.69) | 133.95[a, b, c, d, e] (23.59–449.96) | 186.01 (16.67–658.66) | 148.19[a, b, c, d] (36.49–418.17) | 118.89[a, b, c, d, e] (30.61–459.73) | <0.001 | 0.45 [0.31–0.56] |

Notes:
BV, Beach volleyball; TD, total rally horizontal distance covered; rTD, relative total rally horizontal distance covered TD; PL, player load[TM]; rPL, relative player load[TM]; ePL, explosive player load[TM] (acceleration > 3.5 m/s³); $p$-value and ES represent global statistics for Kruskal-Wallis test.
[a] Significant difference ($p < 0.05$) with Blocker$_{BV}$.
[b] Significant difference ($p < 0.05$) with Defender$_{BV}$.
[c] Significant difference ($p < 0.001$) with Blocker$_{BV}$ and Defender$_{BV}$.
[d] Significant difference ($p < 0.001$) with Blocker.
[e] Significant difference ($p < 0.001$) with Libero.
[f] Significant difference ($p < 0.001$) with Outside.
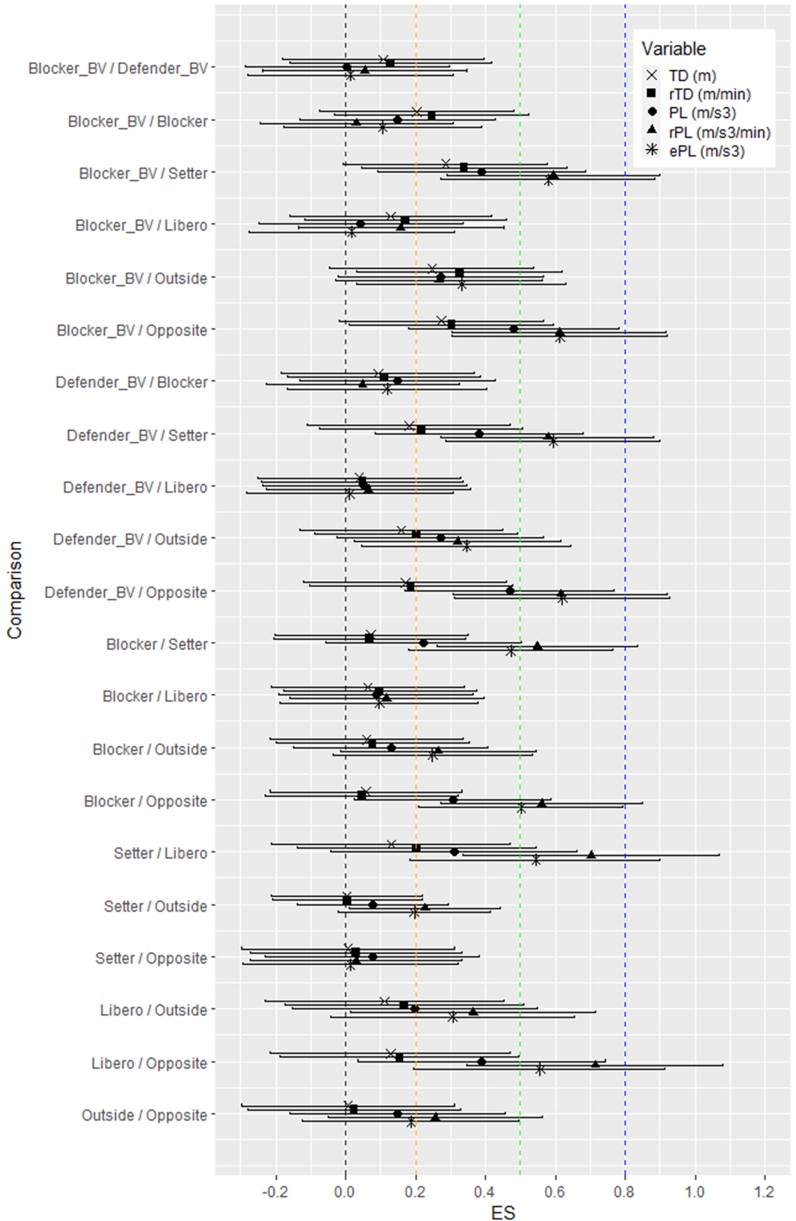

**Figure 2 ES with corresponding confidence intervals for pairwise comparison between player position.**

## DISCUSSION

### Comparison of external load parameters between beach and indoor volleyball

This study found different EL parameters between BV and IV, and between individual player positions. BV showed higher values in distance covered, distance covered per minute and PL parameters, while mean duration of rally play did not differ between two sports. Players achieved different overall work and rest times depending on total match duration. Work-rest ratios differed between female BV (1:5) and IV (1:2) players as

athletes prolong rest times between rallies to recover in BV. Walking or standing in the sand during rest periods in BV also increases the EL (*Sánchez-Moreno et al., 2016*). BV and IV players often stand or walk (0–7 km/h) in between rallies or stand and sit during time-outs or between sets. Energy costs of linear movement on various surfaces were well-studied but we assume it rises significantly when changing direction, making unilateral transitions, or decelerating. Slow movements such as walking on sand also contribute to 1.8 (60–200%) times higher energy cost (J/kg/m) in comparison to movements on a rigid surface. When we consider that work-rest ratio in BV can increase two and a half times as compared to IV, BV players may spend four and a half times more energy during a match just by walking (*Davies & Mackinnon, 2006*). Energy expenditure as players move faster on sand increases from 1.2 to 1.5 times more than on a rigid surface (*Pinnington & Dawson, 2001*; *Zamparo et al., 1992*), and BV players reached up to 20% higher loads when compared to IV players. Not separating rest time from the active play during match performance can distort measurements of player's physical intensity (*Jerome et al., 2023*). While a BV match lasts from 30 min (two sets per match) to 1 h (three-set match), (*Giatsis & Papadopoulou, 2003*) an IV match can last from 1 h in a three-set match to 2 h for a five-set match (*Blair, 2014*). Evaluation of DR depends on how the beginning and end of rallies are determined. Unfortunately, this description is often missing from research and the DR is either calculated from match time and the number of rallies, exported from volleyball statistical software, or customized. In this research, most rallies (80%) in both sports did not exceed 10 s (*Hank et al., 2017*). *Palao et al. (2015)* showed similar but lower DR in BV (from 6.5 ± 4.2 s to 6.9 ± 4.2 s) than our data (7.3 ± 3.5 s). In IV, *Blair (2014)* evaluated a higher value of 10.5 ± 1.1 s. However, *FIVB (2022)* officially reported a value of 8.3 s in female indoor matches, which is comparable with our results (7.50 ± 4.57). Additionally, research has shown longer DR in elite female athletes than males (*FIVB, 2022*; *Inkinen, Häyrinen & Linnamo, 2013*). Our analysis also showed that 85% of TD were shorter than 15 m in both groups while 5% of BV and 8% of IV distances were no longer than 2 m ("*pseudo rallies*" were excluded). Total distances of less than 2 m occurred 1.6 times more often in the IV group. As some rallies are relatively fast and there are more players in IV, some players are not always forced to move through longer, or any, distances (as data showed couple minimum TD around 0.15 m in IV). In a study by *Mroczek et al. (2014)* mean TD reached 10.9 ± 0.9 m during a four-set IV match but was based on male athletes that used a manual marking of the path. The standard deviation of ±0.9 m per rally seems to be remarkably low, especially when compared to the mean distance covered during the match (1,757 ± 462 m), where standard deviation reached almost 30 percent. Mean TD in our study was lower (8.80 m) but had a higher standard deviation (±6.36 m). Importantly, the total match time in both sports is considerably different which is why relative values were evaluated. *Nunes et al. (2020)* revealed a rTD 30 ± 5 m/min, but data were collected throughout the whole match, without separation between active play and rest time. Within higher distances, we also registered higher PL variables in BV. A study on female IV players by *Vlantes & Readdy (2017)* used

an accelerometer to reveal a PL of 417 m/s$^3$ in four sets of collegiate matches and 388 ± 120 m/s$^3$ mean values between three-, four-, and five-set matches, but data was collected throughout the whole game, with no distinction between active play and rest. In terms of acceleration magnitude, we assumed a larger ePL in the IV group as a rigid surface allows players to generate greater propulsion force, but our results reflected a higher volume of accelerations more than 3.5 m/s$^2$ in the BV group. Though unstable forces may limit force propulsion, BV players have more space to accelerate than IV players.

## Comparison of external load parameters based on players' position

From the player position point of view, BV players did not differ between each other in any EL parameter. Beach players had between 11% to 23% higher rTD and PL parameters, while only Libero attained values similar to BV players. This was in agreement with our hypotheses, that BV players show higher EL demands, but not between all player positions. Surprisingly, Blockers attained higher EL than the IV players (excluding Libero). This emphasize the importance of maintaining and developing lateral movement performance and dexterity (*Karalić, Vujmilović & Gerdijan, 2023*), although it may be compromised due to anthropometry.

*João et al. (2021)* showed different ($p < 0.001$) EL between Blocker$_{BV}$ and Defender$_{BV}$ according to several jumps in various heights, time spent in various running speeds, and Peak PL (m/s$^3$/min), but no statistical difference in PL, TD or rTD was found. Individual physical requirements between BV player positions were also found according to jump counts performed by Blocker$_{BV}$ (*Natali et al., 2017*; *Palao et al., 2015*). On the contrary, *Nunes et al. (2020)* affirmed that BV positions did not differ in TD or rTD. When compared to our mean results (between 68 to 79 m/min), rTD reached only 32.9 ± 6.6 m/min. Lower mean values in *João et al. (2021)* are probably skewed by the inclusion of time spent between rallies containing low-velocity walking and standing; in-game time outs and time between sets however were excluded. Additionally, BV consists of short peak-accelerated movement with changes in direction. The reliability of distance measurement by GPS technology may decrease (*Rico-González et al., 2020*), thus video analysis may remain more accurate. Interestingly, rTD was also lower in players assigned the Outside position than in BV players despite being active in all defensive and offensive actions and lacking only setting. Individual performance in offense and defense is closely linked to set distribution and game strategy and thus may influence players' engagement and EL in the match. Volleyball players subjectively perceive attack performance as the most physically demanding in the match (*Junior, 2018*), but some IV positions attack minimally (Setter) or not at all (Libero) during the match. Surprisingly, Libero reached higher values in PL variables among IV positions. Despite Libero's lack of offensive and setting actions, the position's horizontal movement was very active and attained distance and PL values at the levels of BV positions. A study by *Blair (2014)* showed that, throughout the whole match, Libero players spent an average time of 50.6 ± 4.6% and Blockers 71.3 ± 5.6% in a standing position. From the work-rest ratio, Libero reached 29.6% and Blocker 14.6% of work time. Outsides and Setters spent a mean of 35.2% and

31.8% of work time. From the load perspective, it was observed that Libero and Blocker do not participate in all rallies. This contributes to our participation logs when Libero participated in an average of 80% of all rallies and Blockers in 60%. This information is crucial, as both positions' rest times are more than 20% higher than other players.

Blocker$_{BV}$, Defender$_{BV}$, Blocker, Libero, and Outside achieved greater ePL than Setter and Opposite throughout the matches in this study. Acceleration in all axes is manifested not only during vertical jumps in attacks, blocks, and serves, but also in court transitions which are crucial for rapid change of direction on court. A study by *Nunes et al. (2020)* revealed higher accelerations over 3 m/s$^3$ for Defender$_{BV}$ in international matches. As Defender$_{BV}$ is the only infield defense player, almost the whole BV court has to be covered. Total of 51% of game actions by Defender$_{BV}$ happen in a static position with subsequent rapid defensive transition throughout the court (*Jimenez-Olmedo et al., 2017*). All game roles must be performed by both players in BV except for blocking, thus physical demands are high for both players (*Giatsis & Papadopoulou, 2003*). In IV, players perform predominantly in their zones of competency, but at the same time EL also increases when all potential attackers use the net approach in case they get the set, or to confuse the opponent. Among female IV playing positions, Setters usually reach highest mean jump counts, but not the highest mean jump height that is achieved by Opposites (*Skazalski, Whiteley & Bahr, 2018*). In this study, Setter and Opposite reached the lowest values in almost all measured variables. This was not expected in the Setter's case, as players move back and forth from defense to setting. Opposite players are offensive specialists and generally do not engage in receiving; but when in the front row, an Opposite is often active at net defense. They are among the tallest players, reaching greatest vertical jump heights and power outputs among all positions, but they seem to have slower results in horizontal movement tests (*Schaal et al., 2013*).

Elite players in both conditions are highly dependent on physical and anthropometric profiles, but it is also observed in other coherence (*Lidor & Ziv, 2010*). In IV, more offensive player positions like Outside, Blocker, or Opposite develop power for vertical performance while Setters and Libero specialize in horizontal performance and agility (*Schaal et al., 2013*). Understanding the importance of all player positions and filling in the physical gaps on the court when it's not necessarily a part of each players' assigned positions can elevate team effectiveness during a game. For instance, high jump height in Setters can increase net defense success (*Lidor & Ziv, 2010*). Still, it is not adequate to compare Blocker$_{BV}$ to Blocker as there is an absence of fast-serve transition, receive, and set, as well as a relatively different attack approach, and full match-time participation.

According to player position EL differences found in previous research and our data, individual physical requirements and development are necessary throughout the training process, or when transferring career from IV to BV, and *vice versa* (*João et al., 2021*; *Natali et al., 2017*; *Palao et al., 2015*; *Schaal et al., 2013*; *Skazalski, Whiteley & Bahr, 2018*). Physical load of elite sports seems to increase yearly, and it is recommended to monitor and modulate periodization of the workload associated with training, competition, coaches' philosophy, player's attitude, and technical and time availability (*Oliveira et al.,*
*2018*). Training load optimization in volleyball should lead to specific physiological and performance adaptations, thus progressive training stimuli (*García-de-Alcaraz et al., 2020*). This may prevent the effects of over-training and higher risk of injury (*Noce et al., 2008*), especially in youths when contraindications have not yet manifested (*Visnes & Bahr, 2013*).

A limitation of the present study is relatively lower sample size analyzed, as only high performing elite athletes were recruited (*McKay et al., 2021*). Detailed time-consuming data processing and separating active play from inactive were also limitations. Another limitation is the generalizability of kinematic data with other methods such as accelerometers or live motion analysis systems. Research in the future should also focus on movement direction analysis. We encourage future research to distinguish between the work and rest time during EL evaluation, evaluate various levels of competitions, or examine gender and age EL differences in IV and BV. Energy cost and metabolic power are highly dependent on the type of the surface and may affect the IR and ER throughout the match between IV and BV, thus we recommend to focus also on energy costs differences and equivalent distance parameters in future research (*Osgnach et al., 2010*). Gender differences in volleyball sports were mostly found in anthropometrics (*Palao, Manzanares & Valades, 2014*), technical (*Koch & Tilp, 2009*; *Nikos, Karolina & Elissavet, 2009*) or biomechanical (*Fuchs et al., 2019*) research, while TD or PL gender differences have not been examined yet. Considering the inevitable difference in terms of EL in sports with various surface types, it would be recommended to examine the IL parameters like heart rate or rate of perceived exertion as well in the future.

## CONCLUSION

We consider the utility of the kinematic analysis method of employment in the situation, where wearing accelerometer is not allowed, as one of the possible ways to evaluate positional data of movement in all three orthogonal planes during official international volleyball competitions. These findings showed EL differences between beach and indoor volleyball and among player positions. However, distance covered in BV or IV is influenced by individual players' assigned positions, relative player area, and thus their differences in activity may change the overall mean result of the sport. Even on smaller court, BV players attained up to 23 percent higher external loads compared to IV but not amongst one another. Only Libero attained similar values compared to BV players, even though PL is measured by 3D data in all three orthogonal planes of motion. Interestingly, unstable surface did not cause reduced ePL in beach volleyball. It seems that this increased activity in BV is attributed to fewer (two) players on the court, thus a larger area per player. Both players in BV must display all volleyball-specific skills at a high level, while IV players can divide their roles, are often not in constant contact with the ball during rally and can substitute during the match. A detailed evaluation of player movement during official match play is imperative for determining optimal game demands. Various running drills and tracks should consider these overall EL parameters in training periodization. This knowledge would be useful for researchers, coaches, or strength specialists in more specific
individualization and regulation of daily training strategies, yet must be supported by more scientific observations in various conditions.

## Practical applications

Agility drills may focus on maximal acceleration from holding a lower body stance to sprinting 1 to 2 m in IV (approximately 1 to 2 steps) and 2 to 4 m in BV (approximately 2 to 4 steps) before rapid stops or changes in the direction of movement. In most cases (70%), we prefer relatively short, yet intense exercise durations between 5 to 10 s, with total covered distance from 5 to 15 m that are repeated more often, over excessively long (more than 30 s) exercises of moderate intensity. Implementation of short duration (up to 5 s) bouts from 1 to 5 m may cover 10% of all exercises. Approximately 20% of the exercises should have longer duration (15 to 25 s) covering up to 35 m in distance. The number of repetitions may vary and must be monitored due to gradually increasing fatigue and decreasing maximal acceleration and coordination performance. We recommend starting with 10 to 15 min of total exercise, with a 1:2 work-rest ratio in IV and 1:5 in BV. The number of sprints and drills that change direction and distance should be increased up to 20% in BV players, particularly on sand surface, while duration stays the same. Additionally, explosive character of the movement and power exercises should be also Higher EL should be applied not only for the Libero in female IV, but also for the Blocker. This position is often overlooked due to less dexterity and anthropometry in its movements, but it is necessary to develop more because of EL demands during the match. Lateral change of direction up to 4 m should be emphasized even more for the IV Blocker, while BV Blocker may develop linear sprint and deceleration up to 8 m from serve to net defense. We strongly recommend training perception abilities within agility conditioning after players become familiar with the type of exercise involved and the movement technique required to transfer skills with greater specificity as movement in volleyball is mostly initiated from external stimulus. Starts, stops and changes in direction should be based on reaction to trainer signals and eventually from other players. For example, trainers can point right/left or front/back; show colors or numbers associated within a specific movement (rapid stops, squats, digs, jumps, sprints, turns *etc.*), or transition to marked place within drill area.

Besides agility drills, we also recommend EL optimization that simulates play and rally drills. Except for the rallies that last between 4 to 10 s (which often fulfill one or two ball-exchange plays over the net), we emphasize the importance of implementing controlled rallies with a duration between 10 to 20 s in at least 20% of training. Coaches should not ask the players to preserve the rally time by avoiding mistakes or low attack efforts to maintain the longer 15–20 s rally duration. This should be accomplished by keeping players fully motivated with their performance while immediately throwing a new free ball, based on the coach's strategy and after possible mistakes or early scoring, to continue the rally duration without a resting period. Finally, different groups and positional differences within a team outline the importance of individualized athlete monitoring with the aim to optimize sport performance and prevent injury-related factors such as non-functional overreaching. Monitoring individualized players' positions is

important for both: performance optimization (*e.g.*, tapering) and injury prevention (*e.g.*, acute-chronic workload ratio).

## ACKNOWLEDGEMENTS

We thank the Czech Volleyball Federation for their valuable contributions during data collection for this project.

### Funding

This work was supported by the Grant UNCE/HUM/032, and co-funded by Programme for Development of Fields of Study at Charles University number START/SOC/066; SVV under Grant number 260599, by Cooperatio Sport Sciences—Biomedical & Rehabilitation Medicine. The funders had no role in study design, data collection and analysis, decision to publish, or preparation of the manuscript.

### Grant Disclosures

The following grant information was disclosed by the authors:
UNCE/HUM/032.
Programme for Development of Fields of Study at Charles University: START/SOC/066.
SVV under: 260599.
Cooperatio Sport Sciences—Biomedical & Rehabilitation Medicine.

### Competing Interests

The authors declare that they have no competing interests.

### Author Contributions

- Mikulas Hank conceived and designed the experiments, performed the experiments, analyzed the data, prepared figures and/or tables, and approved the final draft.
- Lee Cabell analyzed the data, authored or reviewed drafts of the article, and approved the final draft.
- Frantisek Zahalka conceived and designed the experiments, performed the experiments, authored or reviewed drafts of the article, and approved the final draft.
- Petr Miřátský performed the experiments, analyzed the data, prepared figures and/or tables, and approved the final draft.
- Bohuslav Cabrnoch performed the experiments, prepared figures and/or tables, and approved the final draft.
- Lucia Mala analyzed the data, authored or reviewed drafts of the article, and approved the final draft.
- Tomas Maly conceived and designed the experiments, performed the experiments, authored or reviewed drafts of the article, and approved the final draft.

## Human Ethics

The following information was supplied relating to ethical approvals (*i.e.*, approving body and any reference numbers):

The research was approved by the independent ethics committee of Faculty of Physical Education and Sport at Charles University under the number EC 259/2020.

## Data Availability

The raw measurements are available in the Supplemental File.

## Supplemental Information

Supplemental information for this article can be found online at http://dx.doi.org/10.7717/peerj.16736#supplemental-information.

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

# PeerJ

**Schelling X, Torres L. 2016.** Accelerometer load profiles for basketball-specific drills in elite players. *Journal of Sports Science & Medicine* **15**:585–591.

**Sheppard JM, Cronin JB, Gabbett TJ, McGuigan MR, Etxebarria N, Newton RU. 2008.** Relative importance of strength, power, and anthropometric measures to jump performance of elite volleyball players. *The Journal of Strength & Conditioning Research* **22(3)**:758–765 DOI 10.1519/JSC.0b013e31816a8440.

**Sheppard JM, Gabbett T, Riggs MP. 2012.** *Indoor and beach volleyball players*. Champaign: Human Kinetics Publishers.

**Sheppard JM, Gabbett TJ, Stanganelli LC. 2009.** An analysis of playing positions in elite men's volleyball: considerations for competition demands and physiologic characteristics. *The Journal of Strength and Conditioning Research* **23(6)**:1858–1866 DOI 10.1519/JSC.0b013e3181b45c6a.

**Skazalski C, Whiteley R, Bahr R. 2018.** High jump demands in professional volleyball—large variability exists between players and player positions. *Scandinavian Journal of Medicine & Science in Sports* **28(11)**:2293–2298 DOI 10.1111/sms.13255.

**Smith R. 2006.** Movement in the sand: training implications for beach volleyball. *Strength and Conditioning Journal* **28(5)**:19–21 DOI 10.1519/00126548-200610000-00002.

**Visnes H, Bahr R. 2013.** Training volume and body composition as risk factors for developing jumper's knee among young elite volleyball players. *Scandinavian Journal of Medicine & Science in Sports* **23(5)**:607–613 DOI 10.1111/j.1600-0838.2011.01430.x.

**Vlantes TG, Readdy T. 2017.** Using microsensor technology to quantify match demands in collegiate women's volleyball. *The Journal of Strength and Conditioning Research* **31(12)**:3266–3278 DOI 10.1519/JSC.0000000000002208.

**Wang S, Yuan F. 2022.** On the development of volleyball from the change of volleyball rules. In: *International Conference of Sports Science-AESA*. 29.

**Wik EH, Luteberget LS, Spencer M. 2017.** Activity profiles in international women's team handball using PlayerLoad. *International Journal of Sports Physiology and Performance* **12(7)**:934–942 DOI 10.1123/ijspp.2015-0732.

**Wundersitz DW, Gastin PB, Richter C, Robertson SJ, Netto KJ. 2015.** Validity of a trunk-mounted accelerometer to assess peak accelerations during walking, jogging and running. *European Journal of Sport Science* **15(5)**:382–390 DOI 10.1080/17461391.2014.955131.

**Zamparo P, Perini R, Orizio C, Sacher M, Ferretti G. 1992.** The energy cost of walking or running on sand. *European Journal of Applied Physiology and Occupational Physiology* **65(2)**:183–187 DOI 10.1007/BF00705078.