# Peer review of "Differences in external load among indoor and beach volleyball players during elite matches"

_PeerJ, doi:10.7717/peerj.16736_

## Round 0.1 · original submission · Major Revisions

Dear Authors, the manuscript is interesting and could help trainers to better understand the load of volleyball and how to ameliorate the training.
Unfortunately, the manuscript showed several issues, both methodological and writing. I suggest carefully following all the reviewers' comments, especially reviewer 3, to improve the quality of the work, maybe with the help of a fluent English speaker.

Reviewer 1 ·

Basic reporting

This study analyses and compares the external load between beach volley and indoor volley players, and between player positions. These results could be useful for practitioners and coaches in prescribing training, also considering the difference in performance demands depending on positions. The introduction, materials and methods sections are clear and understandable however I have just some minor issues that need to be addressed before the acceptance :
-line 33: I suggest defining the dependent variable when declaring results in a sentence;
-line 36: why here there isn’t the effect size?
-line 56-59: I suggest a clarification between lines 56 and 59;
-line 89-94: explaining why an indoor volleyball player decides to play beach volley is irrelevant to the purpose of the study.
The error bars in the figures are expressed as standard errors. This makes it more difficult to interpret the variability in the data.

Experimental design

The hypothesis, materials, methods and statistical analysis are well structured, I have no suggestions for them.

Validity of the findings

Results and discussion are clear and understandable however I have some issues that need to be addressed before acceptance :
-lines 189-191: I recommend clarifying lines 189-191;
-lines 241: what do you mean by “more loaded”, I would suggest using a more appropriate term;
-line 245: The last sentence of this line is not linked to the following, so I recommend highlighting this link if it is useful, otherwise delete it.

·

Basic reporting

The work is well written and the aims of the study are well clarified by the authors. The English is clear and understandable to an international audience. The introduction lacks a robust and a detailed description of performance in indoor and beach volleyball in relation to different roles. The bibliography is well edited and up-to-date and the aims of the study are quite clear. The raw data is extensive and I thank you for providing it.

Experimental design

In general the methodology used is clear and well structured. Regarding the forumula used to calculate the PL (Line 138) the reference author has been cited but the formula has not been included, is it possible to include it? The parameters taken into account for the external load are clear and simple to understand, but since volleyball is an intermittent sport, is it possible to calculate the equivalent distance (Osgnach et al. 2010) and perform statistics for this metric?
With regard to statistical analysis, the confidence interval for ES must be included (Batterham AM; and Hopkins WG, 2000). Furthermore, having a small sample, is it possible to use the smallest worthwhile change (Batterham AM; and Hopkins WG, 2000) to highlight differences where no statistically significant differences have emerged? Tables are well structured. The graphs representing the results obtained relative to the ES are absent, can these be included?

Validity of the findings

The data are well represented and find confirmation in the international literature cited by the authors. Could the difference between BV and IV on distance travelled be due to differences in the distance covered by each player in the two different sports? Could you check whether there is any data in the literature on this hypothesis? Possible future research developments are well clarified in the discussion. Possible practical applications for practitioners in relation to the results obtained should be included in the conclusions.

Additional comments

In order of importance, it is necessary to clarify these points:
1. Calculate the equivalent distance and do statistics for this parameter;
2. Enter the confidence interval regarding the ES and the Smallest Worthwhile Change for all outcome;
3. Insert graphs representing the results obtained in relation to ES;
4. Insert SWC results into tables;
5. Check whether the different covered distance could be due to different field sizes in relation to the number of players between the two sports;
6. Enter the formula of PL at line 138;
I congratulate the authors on their work, which is clearly original and well-structured, the result of good scientific planning with solid research hypotheses. Having clarified the points listed above, the article will be ready for publication in our journal.

Reviewer 3 ·

Basic reporting

The article needs extensive revisions to improve the understandability and clarify certain important aspects of the study design (please see the comments regarding the experimental design section).
There are some limitations in the study design and the manuscript lacks some important details (see experimental design for more detailed comments). The introduction also needs revisions to improve its readability and make the primary and secondary goals and rationale of the research clearer to the reader. In particular, the authors should provide a specific background on the topic of interest (e.g., the external load in both beach volley and indoor volleyball and of their comparison or lack of comparison). This could help the reader to follow the information flow immediately. Additionally, certain parts of the text seem to be out of context (i.e., see the comment relative to line 89-94 on additional comments section) and other lack of an appropriate introduction (i.e., see comment relative to line 78 on additional comments section), which makes it difficult to the reader to understand the primary goal of the research and follow the information flow.
Generally, the English language should be improved. For instance, certain parts might be too informal and result difficult to follow for the reader (see additional comments for further details).
About the raw data, they are provided, however there is no recall in the text that states something about it. Furthermore, please explain why columns A and B of the raw dataset are filled just until row 257. My suggestion is to include a legend to make easier the reading.
Figure 1 is not of high quality, while Figure 2 is not easy to understand. Please consider changing Figure 1 and improving Figure 2 clarity.

Experimental design

This research is within the aim and scope of the journal. However, the research question is not well defined and contextualized within the current literature. Indeed, there is no precise statement on which are the primary and secondary aims of the present study.
The methods are not described properly. Specifically, is not mentioned how the sample size was determined and has not been justified why Player Load has been used as an external load index. In this respect, has been shown that Player Load is highly influenced by methodology and by the formula used for its determination. Moreover, the researchers found that Player Load calculations methods present many inconsistencies and lack clear and complete information (Bredt et al., Journal of Human Kinetics, 2020). Please explain why an index which relies on measurements from accelerometers was employed when accelerometers cannot be used and provide a more comprehensive explanation on why you used this method.
About the statistical approach used in this study, is not clear how was managed the multiple testing, hence type 1 error inflation due to multiple comparison, which should be managed in some way. Furthermore, it is not clear which are the dependent and independent variables used in the analyses and how the subgroups analyses (e.g., comparison of the different roles) were performed.

Validity of the findings

The primary aim of the study has not been clearly specified. Indeed, despite there is no question that the article is about the external load demand differences between beach volley and indoor volleyball, the primary aim of the study has not been clearly stated in the manuscript, which also makes the discussion section difficult to follow. Additionally, I would suggest stating the type of study (e.g., observational, prospective, retrospective, etc.) within the manuscript.
Please try to explain more in detail why it is necessary to make a comparison of the external load in different roles in indoor volleyball. In fact, that there is a different demand between different roles is documented both in volleyball (Sheppard et al., Journal of Strength and Conditioning Research, 2009) and in other sports such as soccer (Bangsbo et al., Journal of Sports Sciences, 2006) and does not represent a surprising finding.
In the conclusion section the authors talked about the “optimal game demands”, however, it does not look clear how it is possible to understand the demands of a game without considering the internal load and the perceived exertion of the players relying only on external load metrics. In this respect, is there a reason why you did not detect heart rate and rating of perceived exertion? Given the unavoidable difference from an external load point of view between the two sports, it would have been interesting to make a comparison between internal load parameters as well. Therefore, if this data are not available, this point should at least be discussed in the limitation of the paper.

Additional comments

A further variable has been introduced in the text, the explosive Player Load, which corresponds to the Player Load which considers only movements with an acceleration greater than 3.5 m/s. Although it is interesting to understand the impact on the external load of explosive movements in sports such as volleyball and beach volleyball, there are no supporting references and it is unclear why this cut off (i.e., 3.5 m/s) was chosen.

In general, the discussion is difficult to follow, moving from one topic to another without these being introduced. My suggestion would be to start discussing the primary aim and proceed to simplify the discussion of the results. Furthermore, in the discussion there are too many assumptions that are not accompanied by a possible explanation or reason why they are made.

Please, control and standardize references with each other (i.e., insert capital letters correctly in journal names when necessary).

Line 46-47. Please consider rewriting the sentence in a more fluent way.

Line 48-49. Any reference to support the sentence “Performance analysis often relied on total time or total distance throughout match which may not always reveal characteristics of player position load differences.”?

Line 49. The sentence “If we think about” seems not correct in the context of a professional research paper. It would be better to use impersonal language.

Line 56. Please, specify the term “loads”, despite it seems implicit that you are talking about internal and external load, these concepts have not been yet introduced at that point of the text.

Line 63-65. Please, consider rephrasing this sentence “From increased activity of the lower limb muscles and stabilization in hip, knee, and ankle joints to lower propulsion during force production on sand” to improve comprehensibility of what you mean.

Line 75. Please, consider changing the term “substantiated” with one more appropriate to the meaning.

Line 78. Please, try to expand the concept of the correlation between internal and external load. In fact, it is not sufficient to make a clear link between what you say before (i.e., internal load, metabolic demand etc.) and what you say after (i.e., external load) using just the sentence about correlation.

Line 79. The acronym PL has been introduced in this line, however, has been introduced again in line 136 and not been used, at least always, over the text.

Line 79-82. Please, provide here a more in-depth description of what Player Load is (i.e., how it is usually measured, strength and limitations etc.)

Line 81-82. The sentence “the more the acceleration and movement, the higher the accumulated PL value.” contains all the limitations related to the use of Player Load. Indeed, this metric is not able to consider the magnitude of the acceleration in itself, this means that the accelerations are considered all equal each other, whereas it is known that it is not the case.

Line 89-94. Please, consider justifying better the presence of this sentence about beach volley specialization.

Line 95-98. Please, consider rewriting these sentences. The style (e.g., the use of question marks) comes across as too informal.

Line 98-100. Please, consider rewriting the sentence “Thus, the aim of this study was to examine movement performance of player positions within active playing time in individual rallies of both game types”. It represents one of the most important parts in the introduction section but, written in this way does not come across clearly.

Line 100. Please, clearly define what external load is in your study and from which specific parameters is composed.

Line 107. Please, consider removing “Tour playoff” to start the sentence, it results repetitive whit what you said in line 109 (e.g., Champions play off matches). Additionally, try to decide whether to use words or numbers to describe the number of players. Indeed, in line 105 you utilized words (e.g., Eight elite female BV players) whereas, talking about IV players in line 107 you used numbers (e.g., 14 elite IV players).

Line 137-139. In these lines you cite Nicolella et al., as regard a supposedly standardized formula, however the study you cited conclude that the Player Load is highly affected by methodology and results different between the cartesian calculated Player Load and the Catapult reported Player Load. Thus, does not appear appropriate to use this study to justify the use of this “standardized” formula. Furthermore, using the term “standardized” appears misleading since there is no unambiguousness about which formula should be used to calculate Player Load.

Line 140. Any reference to support the use of this acceleration value as a cut off between non-explosive and explosive movements?

Line 153-154. To open the results section talking about the total duration of the rallies does not seem appropriate. Indeed, total duration of a game, since it is not known players behaviors during this time, should not be considered an external load index.

Line 186-187. I have difficulties understanding the opening sentence of your discussion. Please try to be more specific. Furthermore, what does the p value you reported represent?

Line 189. What do you mean by match demand? Which parameters are involved in this comparison? Please, explain in detail what you mean here.

Line 205-208. Please, consider removing or moving this sentence regarding heart rate. Since the aim of the study is closely related to external load, it is misleading to include (by the way, without having introduced anything about internal load concept into the discussion) a part about heart rate zones. Especially before discussing the results related to Player Load.

Line 206. Please, rewrite the sentence “with a maximum heart rate (HRmax) between 50% and 80%” correctly. Indeed, if I understood correctly what you want to say, it should be written something like “with an heart rate between 50% and 80% of the maximum heart rate”. On the contrary, try to explain what these percentages are representative of.

Line 219. Please, define “considered obscure”.

Line 220. Any reference to support that “considering the gender differences” your results are “comparable”?

Line 225-227. In these lines you tried to compare your results in terms of total distance with results from other that include rest time. Probably you should not include this part, if not, the discussion section is getting full of assumptions without solid foundation. See the comment to line 220 for another example.

Line 231-232. For a comment on “our lower results can be explained by the absence of rest periods.” please refer to what was said for line 220 and 225-227.

Line 232-233. Consider removing the phrase “Our PL and rPL endorses that volleyball is highly dynamic” or please define “highly dynamic”.

Line 233. Please, consider moving the sentence “We encourage future research to distinguish the work and rest time” toward the end of the discussion section, possibly included in a subsection regarding future directions or perspectives.

Line 236-238. Please consider rewriting the sentence in a more understandable way.

Line 240. Where the p value relative to the jump count is coming from?

Line 241. I can understand what you mean by “more loaded” but please try to find a more scientific and professional synonym.

Line 246. What does it mean “peak PL per minute”?

Line 246-267. I found it difficult to follow this part of the discussion. You present too many results from other studies, which would not always be wrong, as long as these are presented with a criterion and in the appropriate manner. Done as it is done in your manuscript, it is confusing for the reader.

Line 305-309. Please, insert a reference to support the data on heart rate you are presenting.

Line 332-333. Please, consider moving the sentence “Research on direction of movement in BV has not been conducted yet, but we assume that it is more balanced than in IV” toward the end of the discussion section, possibly included in a subsection regarding future directions or perspectives.

Line 335-336. Please, consider rewriting the sentence “When transferring from any position from IV, specific skill, agility, aerobic and anaerobic development are necessary” in a more understandable way.

Line 340-344. Please, include references to support what it is written here.

Line 350-351. Please, try to justify why was “successful” the employment of kinematic analysis where no wearing sensor is allowed.

Line 360. What does “real conditions” mean?

Figure 1. Please, consider changing this figure. It does not seem conform to the technical requirements of PeerJ. Additionally, it is not easy for the reader to understand from the Figure how the external load is computed.

Table 1. Is there a reason to repeat the results in both text and tables? From my point of view, it is not necessary to repeat them to avoid redundancy.

Table 2. The table looks confusing to me. In particular, the columns in which you reported ES, p, and post-hoc should be made clearer.

---

## Round 0.2 · Major Revisions

Dear Authors,
The manuscript is improved but the reviewers have highlighted several issues yet. Please, follow carefully the corrections and suggestions provided by the reviewers prior to submitting the revision.
Kind regards.

Reviewer 1 ·

Basic reporting

Dear authors, thank you for responding to my comments and the various changes and rebuttals. However, although many doubts have been resolved, others have emerged in this new version.
There were some typos in the Abstract, for instance, “independed” instead of independent, thus I strongly recommend to make more attention to these details and correcting the several typos present in the paper.
Some words could be deleted and the state remains the same (line 37: cause)

I believe that the introduction should be revised to avoid shifting the focus to other topics unrelated to the comparison between the two disciplines (e.g. lines 100-120 talking about GPS and accelerometers since you are not comparing acquisition methods in your study). If you really want to talk about accelerometers and GPS, this section should not take more than one or two lines, as it is not your topic
Overall I suggest making the introduction more concise.

Line 67: I recommend you to define the word that compounds the acronymous before and then the acronyms.
Line 70-72: I believe that these sentences lack congruence with each other. Furthermore, in the introduction it is written "we have divided the load into internal and external load..." this sentence is taken out of context
Line 74 and lines 79-80: there was a repeated definition of the acronyms IL

Experimental design

Experimental design
The procedures, statistical analysis and results section present several issue that should be addressed.

Line 190-191: Instead of providing a rough estimate of the number of training hours performed by these athletes, I recommend assessing the precise number of training hours per week completed by the athletes participating in this study.

Line 197-198: there is an issue regarding the sample size calculation. Your purpose is to compare the PL between the groups thus your sample size determination shouldn’t be based on rallies but on participants. Furthermore given the low effect size hypothesized and the small sample size I strongly recommend adding in the results section the power of the results (given the likely underpowered results).
Lines 237 to 239: to clarify
Line 250: “Player in the” can be deleted and the result is the same
Line 251: You indicated the Cohen’s d without p-value, although should be present

Line 265-267: to clarify.
Finally, I suggest re-setting the results and the discussion by trying to give an order, starting for example with the comparison between groups and then continuing with the comparison between positions.
Or in another way, I suggest dividing the Results (and discussion) into different sections for each dependent variable.

Validity of the findings

In statistical analysis, you mentioned the calculation of the confidence intervals and the SWC however I don’t find them. Thus, I recommend adding them.

Additional comments

Finally, I strongly recommend making it more concise and sorted to enhance readability

·

Basic reporting

Thank you for providing the requested modifications. However, I have noticed that the confidence interval (CI) for the effect size (ES) is missing from the data. Therefore, I kindly request you to include the CI in both the results and the table. Regarding the smallest worthwhile change, it is essential to ascertain whether the average difference between the two variables exceeds the smallest worthwhile change. Merely stating, for example, "The smallest worthwhile change in the DR was 0.84 s (11%)" (line 258) is insufficient. It is necessary to compare the two variables and incorporate this difference in the table for each variable, accompanied by its respective comment. Providing an interpretation of these data, particularly in cases where the results are non-significant, is crucial. This is especially important in understanding the data better and determining whether these potential minor differences could have an impact on performance, considering that we are dealing with elite athletes who operate at an exceptionally high performance level. Please include this parameter more explicitly in both the text and the tables.

Regarding the graph for the ES, along with its corresponding confidence interval, I apologize for not being clear enough. I am attaching a recently published paper by Salerno P. et al., which presents this type of graph (Figure 4). You can access it at the following link: https://sportperfsci.com/the-use-of-the-parachute-for-improving-sprinting-performance-in-non-elite-sprinters-a-preliminary-study/. I kindly request that you incorporate this type of graph (forest plot) into the manuscript to provide enhanced graphical clarity of the obtained results.

I commend you on the work you have done, and once these minor modifications have been made, the article will be ready for publication in our journal.

Experimental design

No comments to be added

Validity of the findings

No comments to be added

Additional comments

No comments to be added

Reviewer 3 ·

Basic reporting

Compared with the first version, the paper has made significant progress in writing and clarity of the aim and the results. The authors responded appropriately to all requests made in the first round of revisions.
Introduction readability and understandability have been improved, and although there are still some aspects to fix, this section has a good flow. There are no parts entirely out of context in the introduction, which helps the audience better understand the study's background. Generally, attention should still be paid to English in some sentences and the use of punctuation and spacing (see general comments for further details). Moreover, try to make the introduction more straight to the point based on the aim (see more specific comments on the introduction in other parts of the review). Although it is better written compared with the last version, it could still be too long and confusing to the reader.
The discussion section has been improved and provides a more precise flow to the reader. However, please check the punctuation and the correctness of the writing to ensure every sentence is ok and to improve the readability. Furthermore, since the practical applications are fundamental in that paper, they should be written more precisely from a practical point of view.
In fact, at this point, the reader needs clear and precise practical directions rather than more data discussed. Consider rewriting the practical applications to give it a more applied angle toward the real-world setting.

Experimental design

This research is within the aim and scope of the journal. In this version, the research question is better defined and contextualized within the current literature. The study's main aim has been clearly stated so that the reader can better follow the flow of the manuscript.
The methods section has been improved, and the Player Load formula has been inserted in the text. Furthermore, the limitations and the debate around the Player Load concept have been discussed in the text. However, due to causes not under the control of the researchers (i.e., the impossibility of using accelerometers in high-level competitions), it remains that the Player Load metric is used outside the form in which it was designed and how it is usually used.
Please, see comments about lines 166-169 for more precise comments on sample size.

Validity of the findings

The aim of the study now results more straightforward compared to the last version, and it provides a more manageable flow to follow for the audience. It remains to be seen, although a reason for that has been provided, why there was a need to make comparisons between different roles, as pointed out in the first revision since many team sports assume that there are different external loads in different roles. Therefore, it still does not represent a novelty in team sports.
Moreover, since there is a part about training optimization and injury reduction in the introduction, the suggestion is to provide some recommendations based on the data in this respect. If not, please consider removing these aspects from the introduction.

Additional comments

Please, uniform the space between words and before parenthesis (i.e., line 78, there is no space between "games" and the following parenthesis).

Line 66-67. The sentence "We divide the load as an Internal Load (IL) and External Load (EL)" seems not appropriate in the context of a professional research paper. It would be better to use impersonal language.

Line 67. Please, consider better define what you mean for "mechanical stress" or reason to remove it from the factors which are part of the external load.

Line 70-73. Please consider rewriting the sentence "As mentioned by McLaren et al. (2018), the degree of relationship between IL and EL is dependent on its measure and training mode, but it seems they are positively associated" in a more comprehensive way. Although it seems clear what is meant by how it is said it should be improved.

Line 95-96. The sentence "If frequency of accelerations during physical activity increases, PL value also rise" is understandable but could be improved in its fluency.

Line 97. Please, consider changing "is describing" with "describes", it seems more appropriate.

Line 114. Please, try to create a smoother transition at this point.

Line 137. "In conclusion" does not seem appropriate in this part of the text; consider removing or changing it.

Line 148. Please, consider rephrasing the sentence and avoiding the use of "We", which could lead to a writing form not suitable for a scientific paper.

Line 152. Please, consider removing the words "type of the" from "This is a cross-sectional type of the study", it should improve the fluency and sentence correctness.

Line 166-169. The authors specified that 106 rallies were needed using a two-independent group
non-parametric test in the sample size explanation. However, further clarification must be given on the exact test used for sample size calculation because several statistical tests assume that the data are independent of each other. Therefore, considering these rallies are clustered among individuals, hence not independent, the statistical test should be explained, and its assumption should be met.

Line 214. Please, consider changing "Sixty percent" with the numeric format (e.g., 60%) for consistency with other percentages in the text.

Line 291-293. That there were more accelerations above 3.5 m/s in the BV could not represent a novelty. Although the unstable surface may indeed limit the force of propulsion, it is also true that o BV players have more space to accelerate than IV players; this could be a cause that could explain the results in this regard.

Line 297. What does stand for "meterage per minute"? Please, consider explaining or changing it.

Line 375. "Optimalization" sounds uncommon. Please, consider changing it to a different and more common term.

Line 380-382. Please, consider rewriting the sentence "We consider the relatively lower number of subjects and the number of analyzed rallies as a study limit, mainly because of detailed, yet time-consuming data processing and active play time separation" as "A limitation of the present study…". The conclusion section will improve fluency and efficacy.

Line 391-392. The sentence "The raw dataset supporting the conclusions of this research are available for download." does not seem appropriate. Please, consider moving the sentence to a position where discuss the data and not as a last sentence of your discussion.

Line 422. Please, see the comment regarding Line 375.

---

## Round 0.3 · accepted · Accept

I confirm that the author addresses all the reviewers' comments

Reviewer 1 ·

Basic reporting

I congratulate the authors on their efforts to improve the quality of the paper.
As a result of numerous changes, the work has improved considerably.

Experimental design

No comment for this section

Validity of the findings

No comment for this section

Additional comments

The work has a good flow and is clear in its content however I would recommend mainly to make the text more concise and avoid rambling or adding more unsolicited and not useful information in order to assess the hypothesis.

·

Basic reporting

The requested changes were all carried out correctly. The last changes to be carried out relate to lines: 94, 102, 127, 269, 277, 283, 286, 304, 324, 335, 337 where the periods begin with Name of author et al. showed that....

It is necessary to modify these sentences by avoiding beginning this way in order to make the text more free-flowing and pleasing to read.

Experimental design

Nothing to add to the previous editing

Validity of the findings

Nothing to add to the previous editing

Additional comments

I congratulate the authors on their work. Once these minor changes have been made, the work will be ready for publication in our journal.